# Role of the Serotonin Receptor 7 in Brain Plasticity: From Development to Disease

**DOI:** 10.3390/ijms21020505

**Published:** 2020-01-13

**Authors:** Marianna Crispino, Floriana Volpicelli, Carla Perrone-Capano

**Affiliations:** 1Department of Biology, University of Naples Federico II, 80126 Naples, Italy; crispino@unina.it; 2Department of Pharmacy, University of Naples Federico II, 80131 Naples, Italy; perrone@unina.it; 3Institute of Genetics and Biophysics “Adriano Buzzati Traverso”, National Research Council, (CNR), 80131 Naples, Italy

**Keywords:** brain connectivity, brain development, gut-brain axis, neurodevelopmental diseases, neuronal cytoarchitecture, neuroplasticity, regulatory T cells, serotonin (5-HT)

## Abstract

Our knowledge on the plastic functions of the serotonin (5-HT) receptor subtype 7 (5-HT7R) in the brain physiology and pathology have advanced considerably in recent years. A wealth of data show that 5-HT7R is a key player in the establishment and remodeling of neuronal cytoarchitecture during development and in the mature brain, and its dysfunction is linked to neuropsychiatric and neurodevelopmental diseases. The involvement of this receptor in synaptic plasticity is further demonstrated by data showing that its activation allows the rescue of long-term potentiation (LTP) and long-term depression (LTD) deficits in various animal models of neurodevelopmental diseases. In addition, it is becoming clear that the 5-HT7R is involved in inflammatory intestinal diseases, modulates the function of immune cells, and is likely to play a role in the gut-brain axis. In this review, we will mainly focus on recent findings on this receptor’s role in the structural and synaptic plasticity of the mammalian brain, although we will also illustrate novel aspects highlighted in gastrointestinal (GI) tract and immune system.

## 1. Serotonin Overview

### 1.1. Serotonin Metabolism

Brain 5-HT is a neurotransmitter playing a key role in modulating neuronal circuit development and activities. The serotonergic neurons, through their extensive axonal network, are able to reach and influence nearly all the Central Nervous System (CNS) areas. As a consequence, 5-HT regulates a plethora of functions such as sleep and circadian rhythms, mood, memory and reward, emotional behavior, nociception and sensory processing, autonomic responses, and motor activity [1].

Our current understanding of the development, evolution, and function of 5-HT neurotransmission is derived from different model organisms, spanning from invertebrates to vertebrates [2]. It is noteworthy that in all species, the serotonergic network is highly plastic, showing changes in its anatomical organization all through the life of the organisms.

5-HT metabolic pathways, reuptake, and degradation are broadly conserved among multicellular organisms [2]. 5-HT is synthesized from the amino acid tryptophan, which is an essential dietary supplement. Tryptophan is hydroxylated to 5-hydroxytryptophan (5-HTP) by the tryptophan-hydroxylase (TPH)—the rate limiting enzyme for 5-HT biosynthesis. 5-HTP, in turn, is converted in 5-HT by the aromatic L-amino acid decarboxylase. The enzyme TPH has two distinct isoforms encoded by two genes: the *Tph1* is expressed in peripheral tissues and pineal gland, while the *Tph2* is selectively expressed in the CNS and in the enteric neurons of the gut [3]. Studies on TPH -knockout (KO) mice confirmed that the synthesis of 5-HT in the brain is driven by TPH2, whereas the synthesis of 5-HT in peripheral organs is driven by TPH1 [4]. Since 5-HT is unable to cross the blood–brain barrier, at least in adult life, the central and the peripheral serotonergic systems are independently regulated. The synaptic effects of 5-HT are mainly terminated by its reuptake into 5-HT nerve terminals mediated by the 5-HT transporter.

The vast array of brain functions exerted by 5-HT neurotransmission in the CNS is made more complex by the interaction of the 5-HT system with many other classical neurotransmitter systems. Through the activation of serotonergic receptors located on cholinergic, dopaminergic, GABAergic or glutamatergic neurons, 5-HT exerts its effects modulating the neurotransmitter release of these neurons [5,6]. In addition, cotransmission—here defined as the release of more than one classical neurotransmitter by the same neuron—occurs also in 5-HT neurons. Among the cotransmitters released by 5-HT neurons, glutamate [7], and possibly other amino acids [8] were identified. The regulation and functional effects of this neuronal cotransmission are still poorly understood and are the object of intense investigation [9].

### 1.2. Role of Serotonin in Morphological Remodeling of CNS Circuits

In the mammalian brain, 5-HT neurons are among the earliest neurons to be specified during development [10]. They are located in the hindbrain and are grouped in nine raphe nuclei, designated as B1–B9 [11]. Although they are relatively few (about 30,000 in the mouse and 300,000 in humans), they give rise to extensive rostral and caudal axonal projections to the entire CNS, representing the most widely distributed neuronal network in the brain [12].

In addition to its well-established role as a neurotransmitter, 5-HT exerts morphogenic actions on the brain, influencing several neurodevelopmental processes such as neurogenesis, cell migration, axon guidance, dendritogenesis, synaptogenesis and brain wiring [13]. 

Besides the endogenous 5-HT, the brain of the fetus also receives it from the placenta of the mother. Thus, the placenta represents a crucial micro-environment during neurodevelopment, orchestrating a series of complex maternal-fetal interactions. The contribution of this interplay is essential for the correct development of the CNS and for long-term brain functions [14]. Therefore, maternal insults to placental microenvironment may alter embryonic brain development, resulting in prenatal priming of neurodevelopmental disorders [15]. For instance, in mice it has been shown that maternal inflammation results in an upregulation of tryptophan conversion to 5-HT within the placenta, leading to altered serotonergic axonal growth in the fetal forebrain. These results indicate that the level of 5-HT during embryogenesis is critical for proper brain circuit wiring, and open a new perspective for understanding the early origins of neurodevelopmental disorders [16,17,18]. 

The importance of a correct 5-HT level in the brain has been demonstrated by numerous studies on mice models. When the genes involved in 5-HT uptake or degradation are knocked out, the increased 5-HT levels in the brain lead to the altered topographical development of the somatosensory cortex and incorrect cortical interneuron migration [19,20]. On the other hand, the transient disruption of 5-HT signaling, during a restricted period of pre- or postnatal development, using pharmacological (selective serotonin reuptake inhibitor exposure) animal models, leads to long-term behavioral abnormalities, such as increased anxiety in adulthood [21,22]. These animals do not show gross morphological alterations in the CNS suggesting that the lack of cerebral 5-HT may only affect the fine tuning of specific serotonergic circuits. This hypothesis has been recently confirmed using a mouse model in which the enhanced green fluorescent protein is knocked into the Tph2 locus, resulting in lack of brain 5-HT, and allowing the detection of serotonergic system through enhanced fluorescence, independently of 5-HT immunoreactivity. In these mice, the serotonergic innervation was apparently normal in cortex and striatum. On the other hand, mutant adult mice showed a dramatic reduction of serotonergic axon terminal arborization in the diencephalic areas, and a marked serotonergic hyperinnervation in the nucleus accumbens and in the hippocampus [23]. These results demonstrate that brain 5-HT plays a key role in regulating the wiring of the serotonergic system during brain development. Interestingly, the transient silencing of 5-HT transporter expression in neonatal thalamic neurons affects somatosensory barrel architecture through the selective alteration of dendritic structure and trajectory of late postnatal interneuron development in the mouse cortex [24]. Altogether, these findings indicate that perturbing 5-HT levels during critical periods of early development influences later neuronal development through alteration of CNS connectivity that may persist into the adulthood [17,25,26]. Interestingly, recent evidence demonstrated that changes in 5-HT homeostasis affect axonal branch complexity, not only during development but also in adult life [27]. In adult TPH2-conditional KO mice it was shown that the administration of the serotonin precursor 5-hydroxytryptophan was able to re-establish the 5-HT signaling and to rescue defects in serotonergic system organization [27].

Interestingly, in recent elegant experiments that combined chemogenetics and fMRI, it was demonstrated that, in adult mice, the endogenous stimulation of 5-HT-producing neurons does not affect global brain activity but selectively activates specific cortical and subcortical areas. By contrast, the pharmacological increase of 5-HT levels determined widespread fMRI deactivation, possibly reflecting the mixed contribution of central and perivascular constrictive effects [28].

On the whole, findings from genetic mouse models confirm that the level of 5-HT during brain ontogeny is critical for proper CNS circuit wiring, and suggest that alterations in 5-HT signaling during brain development have profound implications for behavior and mental health across the life span. Indeed, a plethora of genetic and pharmacological studies have linked defects of brain 5-HT signaling with psychiatric and neurodevelopmental disorders, such as major depression, anxiety, schizophrenia, obsessive compulsive disorder and Autism Spectrum Disorders (ASD) [17,29,30]. In addition, it is becoming increasingly clear that 5-HT has a crucial role also in the maintenance of mature neuronal circuitry in the brain, opening novel perspectives in rescuing defects of CNS connectivity in the adult. For instance, the potential of 5-HT neurons to remodel their morphology during the entire life is indicated by the well-known capability of 5-HT axons of the adult to regenerate and sprout after lesions [26,31]. 

However, understanding the cellular and molecular mechanisms underlying the effects of 5-HT during brain development, maintenance and dysfunction is challenging, in part due to the existence of at least 14 subtypes of receptors (5-HTRs) grouped in seven distinct classes (from 5-HT1R to 5-HT7R). All 5-HT receptors are broadly distributed in the brain where they display a highly dynamic developmental and region-selective expression pattern and trigger different signaling pathways. The 5-HT receptors are typical G-protein-coupled-receptors with seven transmembrane domains, with the exception of the 5-HT3 receptor, which is a ligand-gated ion channel [32].

## 2. Role of the 5-HT7R in Shaping Neuronal Circuits

### 2.1. The 5-HT7R

The 5-HT7R, the last discovered member of the 5-HTR family [33,34], has always been the subject of intense investigation, due to its high expression in functionally relevant regions of the brain [35,36]. Accordingly, several recent data have elucidated its role in a wide range of physiological functions in the mammalian CNS and also in peripheral organs [37]. Interestingly, emerging findings indicate that 5-HT7R is involved in brain plasticity, being one of the players contributing not only to shape brain networks during development but also to remodel neuronal wiring in the mature brain, thus controlling higher cognitive functions (see Section 2.2 and Section 2.3). Therefore, this receptor is currently considered as potential target for the treatment of several neuropsychiatric and neurodevelopmental disorders, (as discussed in Section 3), also in view of the fact that its ligands have a wide range of neuropharmacological effects [38,39].

In the mammalian CNS, the 5-HT7R is mainly expressed in the spinal cord, thalamus, hypothalamus, hippocampus, prefrontal cortex, striatal complex, amygdala and in the Purkinje neurons of the cerebellum [40,41]. This wide distribution reflects the numerous functions in which the receptor is involved, such as circadian rhythms, sleep-wake cycle, thermoregulation, learning and memory processing, and nociception [37]. 

In mammals, this receptor exhibits a number of functional splice variants due to the presence of introns in the 5-HT7R gene and to alternative splicing. The splice variants of the receptor, named 5-HT7(a), (b), (c) in rodents, and 5-HT7(a), (b), (d) in humans [36,42,43], do not show significant differences in localization, ligand binding affinities, and activation of adenylate cyclase [36]. To date, the only functional difference between the splice variants is that the human 5-HT7(d) isoform displays a different pattern of receptor internalization compared to the other isoforms [44].

The 5-HT7R is a G protein-coupled receptor, that activates at least two different signaling pathways. The classical pathway relies on the activation of Gαs and the consequent stimulation of adenylate cyclase, leading to an increase in cyclic adenosine monophosphate (cAMP). The latter activates protein kinase A (PKA), that in turn phosphorylates various proteins such as the mitogen-activated protein kinase and extracellular signal-regulated kinases (ERK) [39]. 

Another 5-HT7R pathway depends on the activation of Gα_12_, that in turn triggers stimulation of Rho GTPases, Cdc42 and RhoA; these intracellular signaling proteins, critical for the regulation of cytoskeleton organization, lead to morphological modifications of fibroblasts and neurons [45]. 

5-HT7R signaling also involves changes in intracellular Ca^2+^ concentration and Ca^2+^/calmodulin pathways [46,47], as well as PKA independent mechanisms which include exchange protein directly activated by cAMP (EPAC) signaling [48].

5-HT receptor signaling has been recently shown to also depend on their oligomerization. In particular the 5-HT7R can form homodimers, as well as heterodimers with 5-HT1AR [49]. The latter, when is in a monomeric conformation, causes a decrease in cAMP concentration through activation of the Gi. Heterodimerization with 5-HT7R inhibits the 5-HT1AR cAMP signaling pathway, while homodimerization of both receptors do not influence the respective cAMP pathways. These findings suggest that oligomerization of G-protein-coupled-receptors may have profound functional consequences on their downstream signaling, thus triggering cellular and developmental-specific regulatory effects.

### 2.2. Role of the 5-HT7R in Shaping Neuronal Circuits during Development

The influence of the 5-HT7R on neuronal morphology has stimulated interest in studying its potential role in the establishment and maintenance of brain connectivity and in synaptic plasticity. The availability of selective agonists and antagonists, as well as that of genetically modified mice lacking the 5-HT7R, has shed light on the physio-pathological role of this receptor [39,50,51]. By using rodents’ primary cultures of hippocampal neurons and various 5-HT7R agonists in combination with selective antagonists, it was consistently shown that the pharmacological stimulation of the endogenous 5-HT7R promotes a pronounced extension of neurite length [48,52,53]. The morphogenic effects of 5-HT7R stimulation have also been demonstrated in cultured neurons from additional embryonic forebrain areas, such as the striatum and the cortex [54,55] (Figure 1). Neurite elongation was shown to rely on *de novo* protein synthesis and multiple signaling systems, such as ERK, Cdk5, the RhoGTPase Cdc42 and mTOR. These pathways converge to promote the reorganization of the neuronal cytoskeleton through qualitative and quantitative changes of selected proteins, such as microtubule-associated proteins and cofilin [54,56]. In hippocampal neurons, it has been demonstrated that 5-HT7R finely modulates the NMDA receptors activity [57,58]. Furthermore, 5-HT7R activation increases phosphorylation of the GluA1 AMPA receptor subunit and AMPA receptor-mediated neurotransmission in the hippocampus [59,60]. Consistent with these findings, 5-HT7R-KO mice display reduced LTP in the hippocampus [61].

Chronic stimulation of the 5-HT7R/Gα_12_ signaling pathway promotes dendritic spine formation, enhances basal neuronal excitability, and modulates LTP in organotypic slices preparation from the hippocampus of juvenile mice. Interestingly, 5-HT7R stimulation does not affect neuronal morphology, synaptogenesis, and synaptic plasticity in hippocampal slices from adult animals, probably due to decreased hippocampal expression of the 5-HT7R during later postnatal stages [62]. It has been recently hypothesized that this decline could be due to the simultaneous upregulation of the microRNA (miR)-29a in the developing hippocampus. Indeed 5-HT7R mRNA is downregulated by the miR-29a in cultured hippocampal neurons, and miR-29a overexpression impairs the 5-HT7R-dependent neurite elongation [63].

Neuronal remodeling is highly influenced by the extracellular matrix. Accordingly, it has been shown that the physical interaction between the 5-HT7R and the hyaluronan receptor CD44, a main component of the extracellular matrix, plays a crucial role in synaptic remodeling. Briefly, stimulation of the 5-HT7R increases the activity of the metalloproteinase MMP-9, which, in turn, cleaves the extracellular domain of CD44. This signaling cascade promotes detachment from the extracellular matrix, thus triggering dendritic spine elongation in the hippocampal neurons of the mice [64].

In accordance with the influence of the 5-HT7R signaling pathways in remodeling developing forebrain neuron morphology, it was shown that prolonged stimulation of this receptor and the downstream activation of Cdk5 and Cdc42 increased the density of filopodia-like dendritic spines and synaptogenesis in cultured striatal and cortical neurons [65]. The crucial role of 5-HT7R in shaping developing synapses (Figure 1) was confirmed by the pharmacological inactivation of the receptor as well as through the analysis of early postnatal neurons isolated from 5-HT7R-deficient mice. It is noteworthy that, when 5-HT7R was blocked pharmacologically, and in 5-HT7R-KO neurons, the number of dendritic spines decreased, suggesting that constitutive receptor activity is critically involved in dendritic spinogenesis. From this point of view, a detailed analysis of dendritic spine shape and density in the brain of 5-HT7R-KO mice at various ages would be crucial to assess the physiological effects of this receptor on neuronal cytoarchitecture. 

The involvement of 5-HT7R in spinogenesis and synaptogenesis—together with the demonstration that its activation is able to stimulate protein synthesis-dependent neurite elongation, as well as axonal elongation [54,56]—suggests the intriguing possibility that the activation of this receptor may be linked to the axonal and synaptic system of protein synthesis. The local system of protein synthesis has been demonstrated to play a crucial role in synaptic plasticity—although its regulatory mechanisms are only partially understood [66,67,68]—and 5-HT7R and its related pathways are good candidates to be part of this system.

### 2.3. Role of the 5-HT7R in Remodeling Neuronal Circuits in Adults

Neuronal circuits remain able to reorganize in response to experience well into adulthood, continuing to exhibit robust plasticity along the entire life [69]. Consistently, the action of 5-HT7R on the modulation of neuronal plasticity is not restricted to embryonic and early postnatal development, but can also occur in later developmental stages and in adulthood (Figure 1).

Interestingly, it was shown that selective pharmacological stimulation of 5-HT7R during adolescence determines its persistent upregulation in adult rat forebrain areas [70]. Likewise, it has been hypothesized that 5-HT7R may underlie the persistent structural rearrangements of the brain reward pathways occurring during postnatal development, following exposure to methylphenidate, the elective drug for the treatment of Attention Deficit Hyperactivity Disorder [71]. Accordingly, stimulation of the 5-HT7R in adolescent rats leads to increased dendritic arborization in the nucleus accumbens—a limbic area involved in reward—as well as increased functional connectivity in different forebrain networks likely to be involved in anxiety-related behavior [72]. Changes in dendritic spine formation, turnover and shape occur during the entire life span in response to stimuli that trigger long-term alterations in synaptic efficacy, such as LTP and LTD [73,74,75]. Consistently, it has been shown that the activation of 5-HT7R in hippocampal slices from wild type mice (as well as in Fragile X Syndrome mice, see next paragraph) reverses LTD mediated by metabotropic glutamate receptors (mGluR-LTD), a form of plasticity playing a crucial role in cognition and in behavioral flexibility [59]. Moreover, the acute in vivo administration of a selective 5-HTR7 agonist improved cognitive performance in mice [76]. These results are consistent with the hypothesis that long-term changes of synaptic plasticity, which are a substrate of learning and memory formation, lead to neural network rewiring (Figure 1). Accordingly, the 5-HT7R-KO mice exhibit reduced hippocampal LTP, and specific impairments in contextual learning, seeking behavior and allocentric spatial memory [61,77]. 

Interestingly, the expression level of 5-HT7R in the hippocampal CA3 region, an area of the brain involved in allocentric navigation, decreases with age [78], suggesting that the spatial memory deficits associated with aging could be attributed to decreased 5-HT7R activity in this region of the brain. Conversely, another group reported that hippocampal expression of 5-HT7R does not change with age, but exhibits 24 h rhythms [79]. This observation should be taken into account in the interpretation of previous findings, as well as in planning future experiments. Several other studies have produced contradictory results related to the involvement of 5-HT7R in memory and attention-related processes [80,81], probably due to experimental differences (animal strain, behavioral tests, compounds and doses, route of administration, etc.). In conclusion, although the role of this receptor on cognitive functions needs to be fully elucidated, it is clear that it modulates various aspects of learning and memory processes. 

Interestingly, the 5-HT7R is also involved in bidirectional modulation of cerebellar synaptic plasticity, since its activation induces LTD at the parallel fiber-Purkinje cell synapse, whereas it blocks LTP induced by parallel fiber stimulation [41]. These results suggest that the receptor might be involved in motor learning, a cognitive function depending on the activity of cerebellar circuits [82].

Altogether, these findings strongly suggest that the 5-HT7R plays a role in modulating synaptic plasticity and neuronal connectivity in both developing and mature brain circuits, although the molecular and cellular mechanisms underlying this modulation are only partially understood (Figure 1).

## 3. The 5-HT7R and Neurological Diseases

Numerous brain disorders, such as ASD, cognitive and mood dysfunctions, schizophrenia, depression, anxiety, impulsivity, epilepsy, migraine and neuropathic pain show altered 5-HT7R-mediated signaling [38]. The potential involvement of 5-HT7R in most of these diseases was discovered studying the effects of a broad range of antidepressant and antipsychotic drugs that interact with the receptor, displaying high affinity [39]. Recently, the importance of 5-HT7R modulation was brought to the attention of psychiatric and pharmacological communities, since a novel very effective and atypical mood-stabilizing antipsychotic drug, lurasidone, predominantly blocks this receptor. This drug also acts as agonist of the 5-HT1AR. Experiments on animal models indicate that chronic treatment with lurasidone enhances 5-HT transmission in dorsal raphe nuclei by coordinated 5-HT1AR agonism and 5-HT7R antagonism through modulation of GABAergic and glutamatergic pathways, thus contributing to the augmentation of the drug’s antidepressive effects [83,84].

In line with the possible involvement of 5-HT7R in the mechanisms of action of antidepressants, genome-wide association studies in humans have suggested a relationship between 5-HTR7 genetic polymorphisms and schizophrenia [85]. Likewise, a very recent work showed that one single nucleotide polymorphism located in the promoter region of the 5-HT7R gene is associated with a better response to two antidepressants, paroxetine and fluoxetine, that are selective 5-HT reuptake inhibitors. These data provide novel pharmacogenomic evidence to support the role of 5-HT7R in antidepressant response [86].

However, the pharmacological and genetic manipulation of 5-HT7R in animal models of depression, anxiety and schizophrenia has often given inconsistent or conflicting results. Experimental differences (for instance animals’ strain, behavioral tests, drugs and their doses, route of administration), as well as the use of non-selective drugs targeting other receptors in addition to the 5-HT7R, might account for these mixed results. In addition, the interpretation of behavioral data from 5-HT7R-KO mice is complicated by the indirect effects of the missing gene, such as changes in developmental processes or dysregulation of compensatory genes and pathways [87]. Very recently, mixed outcomes on various behavioral assays for anxiety, depression and psychosis, performed on mice treated with two selective 5-HT7R antagonists [88], and on 5-HT7R-KO mice [89], raised doubts on the role played by the receptor in these neuropsychiatric diseases. Ultimately, the available data suggest that additional research will be required to further evaluate and dissect the contribution of this receptor in anxiety/depression and schizophrenia, and its potential involvement for the treatment of these neuropsychiatric diseases.

Conversely, compelling evidence strongly suggests that the 5-HT7R is involved in CNS disorders characterized by intellectual disabilities and cognitive impairment, such as Rett syndrome (RTT) and Fragile X syndrome (FXS; Figure 1). These diseases belong to ASD, a heterogeneous group of neurodevelopmental disorders characterized by impaired social interaction and communication, repetitive and stereotyped behaviors, often accompanied by cognitive defects [90]. Growing evidence indicates that the brain 5-HT neurotransmission system is altered in ASD patients, and in various animal models of the disease [91,92]. For instance, mice lacking brain 5-HT, in addition to several abnormal phenotypes (growth retardation, high aggressive behavior, maternal neglect), show selective deficits resembling ASD’s symptoms, including impairment in social interactions and repetitive behavior [3,4]. Various pharmacological studies are providing evidence that targeting 5-HTRs has the potential to treat the core symptoms of ASD and associated intellectual disabilities [93,94]. Recent evidence in animal models suggests that, among other subtypes, the 5-HT7R might be one of the players involved in ASD (see below). In line with this hypothesis, the only two drugs that to date are approved for the treatment of behavioral manifestations of ASD, risperidone and aripiprazole, are 5-HT7R antagonists [95], although their efficacy may be attributed also to their interactions with other receptors. Indeed, none of the approved ASD drugs are highly selective for 5-HT7R, hampering our understanding of its potential as a target for pharmacological treatment of ASD in humans. Nevertheless, brain-permeant and selective agonists of the 5-HT7R have been successfully employed to rescue ASD dysfunctions in animal models of FXS, RTT and CDKL5 Deficiency (CDD).

FXS mice exhibit cognitive impairment and stereotyped behavior, accompanied by altered morphology and density of dendritic spines in the forebrain, alongside synapse malfunctioning in the hippocampus, with the abnormal enhancement of mGluR-LTD. The activation of 5-HT7R by a selective brain-permeant agonist in hippocampal slices from FXS mice is able to correct excessive mGluR-LTD through activation of cAMP/PKA pathway, bringing it back to its physiological level and thereby restoring its synaptic plasticity. Noteworthy, acute in vivo administration of the agonist rescues learning and autistic-like behavior in 3/4 months-old FXS mice [76]. 

Beneficial effects of the same agonist, chronically administered, were also observed in adult mouse models of RTT. This syndrome is a severe X-linked neurological disorder characterized by deficits in autonomic, cognitive, motor functions and autistic features. In vivo systemic repeated stimulation of 5-HT7R with a selective brain-permeant agonist was able to improve cognitive and motor coordination deficits, as well as spatial memory and synaptic plasticity in RTT mice. 5-HT7R stimulation also restored the normal level of key molecules regulating actin cytoskeleton dynamics, such as Rho GTPases and mTOR signaling pathways that showed altered expression levels in the hippocampus of RTT mice [96,97]. The 5-HT7R-mediated neurobehavioral and molecular changes were still present 2 months after the last injection, suggesting long-lasting, beneficial effects on RTT-related impairments. Subsequent studies uncovered functional alterations of brain mitochondria in RTT mouse models, that were rescued by the chronic pharmacological stimulation of the 5-HT7R [98,99]. Similar promising preclinical results have been recently obtained in a mouse model of CDD, a rare neurodevelopmental syndrome characterized by severe neurobehavioral and motor deficits and stereotyped movements [100].

Altogether, the above findings provide compelling evidence that 5-HT7R stimulation exerts a widespread beneficial effect on behavioral and molecular symptomatology in various mouse models of neurodevelopmental disorders, in particular those belonging to ASD (Figure 1). Moreover, these results have important therapeutic implications, indicating that it is possible to reverse severe behavioral and molecular deficits in the animal models by pharmacological treatment at adult age. Intriguingly, all these diseases are accompanied by the alteration of dendritic spines in forebrain areas involved in higher cognitive functions, suggesting altered connectivity. Although data on the 5-HT7R-dependent remodeling of dendritic spines in the ASD animal models are still missing, it is possible to hypothesize that the activation of 5-HT7R may also promote structural rearrangements of neural circuits in the adult brain, that in turn might underlie the rescue of long-term synaptic plasticity. 

## 4. The 5-HT7R in the Gut and in the Immune System

Despite the vast repertoire of neurodevelopmental, behavioral and cognitive processes modulated by the brain 5-HT, only ~5% of the total body content of 5-HT is located in the CNS, while the remaining part is synthesized and stored in peripheral tissues.

Outside the CNS, the vast majority of 5-HT is found in the gastrointestinal (GI) epithelium, where it is mainly produced by enterochromaffin (EC) cells of the gut mucosa and only in small quantity by neurons of the Enteric Nervous System and by the resident gut microbiota. 5-HT released by EC cells is actively taken up and stored by blood platelets, and released upon their activation, modifying vascular smooth muscle tone and a variety of other functions controlled by peripheral organs [101]. In the gut, in physiological conditions, 5-HT made by EC cells and enteric neurons act synergistically to regulate the intestinal functions, such as motility, sensation, and secretion, whereas alteration in the 5-HT metabolism is associated with various diseases of the GI tract (see below).

Peripheral 5-HT is also a potent immune system modulator and can affect various immune cells through its receptors. In addition, 5-HT is synthesized and released by some cells of the immune system (T limphocytes and mast cells), expanding the range of tissues involved in its signaling [102]. Peripheral 5-HT7R expression roughly mirrors peripheral 5-HT distribution, since it has been observed in the GI tract, as well as in the peripheral organs (kidney, liver, pancreas, spleen, and stomach), and in cells of the immune system [103].

Here, we briefly review recent studies showing that 5-HT7R plays a crucial role in generation/perpetuation of intestinal inflammation, and in immune cell activation. The immune system is known to play an important intermediary role in the dynamic equilibrium between the CNS and the GI tract [104]. Therefore, it is intriguing to hypothesize the involvement of 5-HT7R in bidirectional communication between the brain and gut, possibly mediated by the immune system (Figure 2). As a key element of this axis, 5-HT signaling may link emotional and cognitive areas of the brain with peripheral gut activity. Interestingly, recent findings suggest that the alteration of this two-way serotonergic system of communication between brain and gut may play a role in the pathogenesis of various diseases, including ASD [105]. Moreover, it is becoming increasingly evident that the resident gut microbiota, that produce tryptophan and 5-HT, is a critical component of the gut-brain communication, modulating brain development and behavioral responses [106,107]. The 5-HT effects in this complex microbiota-gut-brain communication are mediated by 5-HT receptors and, among other subtypes, the 5-HT7R is a very interesting candidate, being expressed both in the gut and in the brain. 

Various findings suggest that 5-HT7R may have a crucial role in the pathogenesis of inflammatory disorders affecting the GI tract, such as Inflammatory Bowel Disease (IBD), which includes ulcerative colitis and Crohn’s Disease. IBD is characterized by activation of the immune cells accompanied by their infiltration in the gut and inflammation of the GI tract, leading to profound alteration of the GI function and dysfunctions of 5-HT signaling [108]. 

Interestingly, genetic or pharmacological silencing of 5-HT7R in mouse models of ulcerative colitis reduced the severity of intestinal inflammation and decreased the production of inflammatory markers by GI dendritic cells. These antigen-presenting cells initiate adaptive immune responses upon inflammation [109]. These results, indicating that 5-HT7R inhibition reduces inflammation symptoms in gut inflammatory disorders, are in contrast with other findings. Indeed, Guseva et al. [110] reported that pharmacological blockade or genetic ablation of 5-HT7R resulted in increased severity of symptoms in both acute and chronic mouse models of colitis, whereas receptor stimulation produced an anti-inflammatory effect. In addition, expression of 5-HT7R significantly increased after induction of colitis in mice and in inflamed intestinal dendritic cells of patients with Crohn’s disease.

Novel epigenetic mechanisms regulating 5-HT7R expression have been recently highlighted by studies on animal models and patients with Irritable Bowel Syndrome (IBS), a functional GI disorder often associated to visceral hyperalgesia without inflammatory processes. It was shown that miR-29a modulates visceral hypersensitivity in a mouse model of IBS by directly targeting the 5-HT7R and downregulating its expression. The authors found that intestinal tissues from mice and patients with IBS displayed increased levels of miR-29a and reduced levels of 5-HT7R. Consistently, in mice with IBS, when miR-29a was knocked-out, 5-HT7R was overexpressed and intestinal hyperalgesia was attenuated [111]. These findings suggest that colon hypersensitivity may be mediated by the endogenous interaction between miRNA-29a and 5-HT7R, offering a potential promising therapeutic approach for reversing abdominal pain in IBS patients.

The discrepancies on the role of 5-HT7R in gut disorders may depend on the notable differences in animal models and experimental design. Moreover, it is possible that immune cells are differentially recruited depending on the experimental model of induced intestinal damage and human gut diseases, and that their production of pro-inflammatory and anti-inflammatory cytokines is modulated by the level of 5-HT and by 5-HT7R expression. Indeed, it has been demonstrated that the 5-HT7R–Cdc42-mediated signaling regulates dendritic cell morphology and enhances chemotactic motility [112]. Likewise, a prominent role of 5-HT7R in regulating endothelial cell migration has been identified, suggesting that this receptor is a potential modulator of physiological and pathophysiological processes involving cell migration, adhesion [113] and inflammatory fibrotic infiltration [103,114]. Thus, 5-HT7R-dependent activation and the migration of dendritic cells might be significantly different in various intestinal inflammatory diseases, accounting—at least in part—for conflicting results on the role of 5-HT7R in the pathogenesis of gut inflammatory disorders. As a conclusion, although it is clear that 5-HT7R can influence gut inflammation, additional studies are required to precisely understand 5-HT7R function and dysfunction in the intestine. 

Nevertheless, these findings highlight the involvement of the 5-HT7R in the physiology and pathology of the immune system. Indeed, it is well known that 5-HT plays a key role in inflammation, immunity and immunomodulatory diseases and that almost all immune cells express at least one 5-HT receptor, both in rodents and humans. 5-HT7R expression has been detected in lymphoid progenitor cells, mast cells, monocytes, macrophages, dendritic cells and T lymphocytes [103]. This receptor is also expressed by microglia, the brain resident macrophages, and its stimulation in human microglial cell lines leads to increased IL-6 expression—a proinflammatory cytochine [115]. The stimulation of 5-HT7R by 5-HT enhances the proliferation and activation of mouse naive T cells through ERK signaling [116]. Likewise, in human macrophages, the anti-inflammatory and pro-fibrotic activity of 5-HT is primarily mediated by 5-HT7R-PKA pathway [114]. Notably, it has been recently shown that brain regulatory T (Treg) cells are distinct from those of other tissues, since they express unique genes related to the Nervous System, including the 5-HT7R. The specific features and functions of brain Treg cells are poorly understood, because their number is very low in the brain under normal conditions. Conversely, a large number of Treg cells infiltrate the mouse brain during the chronic phase of ischemic stroke, suppressing astrogliosis and potentiating neurological recovery [117]. Brain Treg cells, but not splenic Treg cells, respond to 5-HT by increased proliferation and this response was blocked by a selective antagonist of the 5-HT7R. Notably, 5-HT7R-deficient Treg cells do not expand correctly into the brain and do not promote neurological recovery after ischemic stroke. These findings demonstrated that 5-HT7R play a specialized role in Treg cells and suggest that the 5-HT signaling mediated by Treg cells might represent one of the mechanisms that contribute to the cross talk between immune system and brain inflammation. 

Altogether, the reported findings on the 5-HT7R signaling in the CNS, GI tract and immune system suggest the involvement of this receptor in inflammatory and immune-mediated disorders affecting the gut and brain (Figure 2). For instance, as mentioned above, two very recent studies showed that 5-HT7R is a direct target of miR-29a in the brain, as well as in the intestine [63,111], suggesting that this miRNA might modulate 5-HT7R expression in both tissues in a coordinated way. 

## 5. Conclusions and Future Perspectives

The modulation of 5-HTR7 expression using pharmacological and genetic tools, coupled to cellular, molecular, electrophysiological and behavioral approaches, has greatly increased our knowledge on the functions of this receptor in the brain, as well as in other organs.

The results highlighted here indicate that 5-HT7R is an important player involved in the modulation of synaptic and structural plasticity in both developing and mature brain circuits. However, the detailed molecular mechanisms and signaling pathways underlying 5-HT7R morphogenic effects are still objects of intense investigation. We believe that conditional KO mouse models are able to silence/overexpress the 5-HT7R gene during selected age windows, and—in specific cell types—would help to more accurately define the contribution of the 5-HT7R to brain physiology.

Altered 5-HT7R-mediated signaling is involved in numerous brain diseases. In particular, compelling evidence indicates that 5-HT7R stimulation reverts behavioral, molecular and functional deficits in various animal models of neurodevelopmental disorders. These 5-HT7R beneficial effects, at least in RTT, might operate through the rescue of the mitochondrial dysfunctions associated with the disease. Thus, it would be of great interest to deepen our understanding on the mechanisms underlying the regulatory effects of 5-HT7R on mitochondrial function.

Finally, we briefly discussed recent findings highlighting a crucial role of the 5-HT7R in intestinal inflammation and immune cell activation, and suggested its possible involvement in the complex interaction between the brain, immune cells and gut.

Importantly, the findings described here open new avenues in the development of selective drugs targeting 5-HT7R as novel potential therapeutic agents in many diseases so far considered incurable.

## Figures and Tables

**Figure 1 ijms-21-00505-f001:**
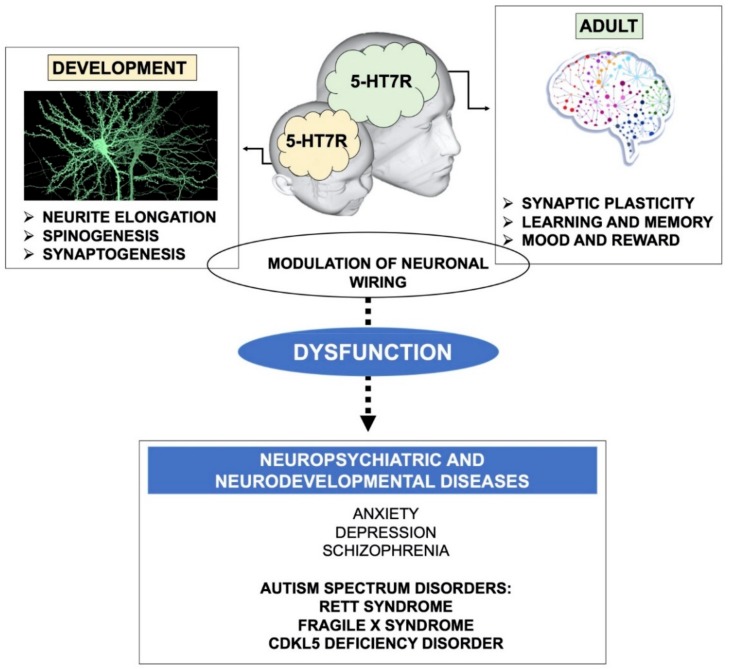
Schematic drawing illustrating the role of the 5-HT7R in brain plasticity and connectivity. During development, the 5-HT7R contributes to proper neuronal wiring through the stimulation of neurite elongation, growth and maturation of dendritic spines, and synaptogenesis. During adulthood, the 5-HT7R signaling stimulates synaptic plasticity (LTP, LTD and structural remodeling of neuronal connections), which in turn affects many physiological functions, such as learning, memory, mood and reward. Dysregulated 5-HT7R signaling was demonstrated in neuropsychiatric and neurodevelopmental diseases characterized by altered brain connectivity. Notably, 5-HT7R stimulation exerts a widespread beneficial effect on behavioral and molecular alterations in various mouse models of Autism Spectrum Disorders (highlighted in bold).

**Figure 2 ijms-21-00505-f002:**
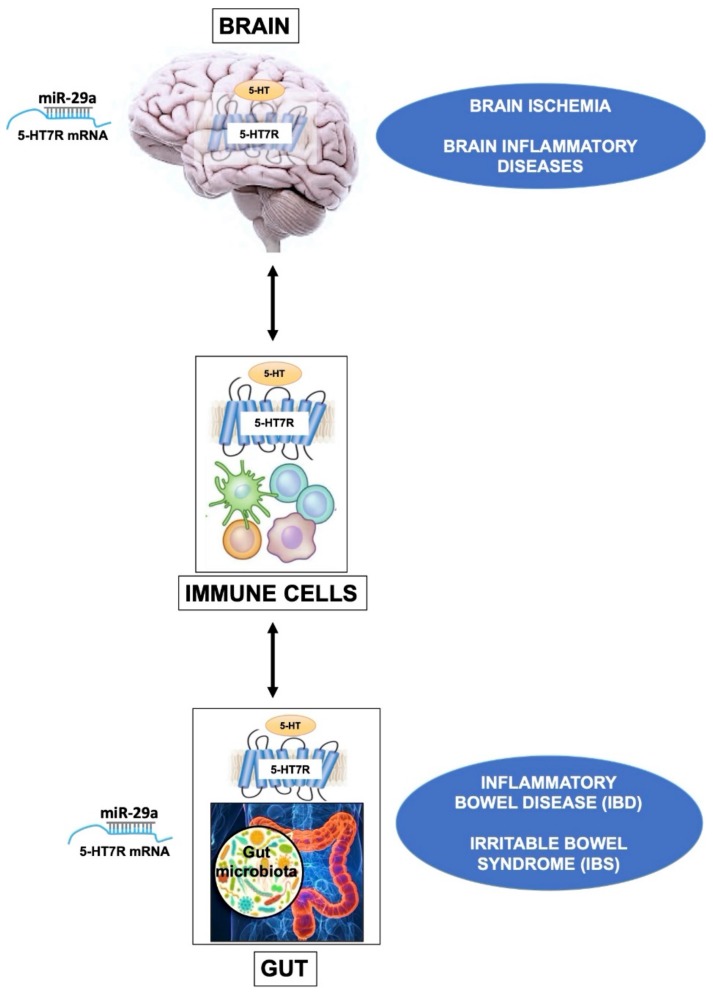
The drawing outlines the influence of 5-HT7R in brain, gut and immune cells. In the CNS, the 5-HT7R expressed by brain Treg cells displays specialized functions that contribute to their proliferation after ischemic stroke, promoting neurological recovery. A similar mechanism is likely to occur also in other neuroinflammatory diseases. In immune cells, the basal activity of 5-HT7R is likely to play a key role in the maintenance of homeostasis. In the gut, alteration of the 5-HT metabolism is associated with various diseases such as Inflammatory Bowel Disease (IBD), and Irritable Bowel Syndrome (IBS). In these dysfunctions the activation of 5-HT7R on dendritic cells modulates their immune response, resulting in either beneficial or detrimental effects, depending on experimental models. It is worth to underline that the expression level of 5-HT7R is epigenetically modulated by the microRNA-29a (miR-29a) in the brain as well as in the gut.

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
