# Peer review of "Role of the Serotonin Receptor 7 in Brain Plasticity: From Development to Disease"

_ijms, 2020, doi:10.3390/ijms21020505_

Round 1

Reviewer 1 Report

This review importantly introduces understanding of the role of serotonin receptor 7 signalling pathway in circuit formation during development as well as leading neurological disorders such as ASD. Additionally, the review is well written and tightly argued, references appropriate. My support for publication in International Journal of Molecular Sciences is very strong.

Author Response

Response to Reviewer 1 Comments

This review importantly introduces understanding of the role of serotonin receptor 7 signalling pathway in circuit formation during development as well as leading neurological disorders such as ASD. Additionally, the review is well written and tightly argued, references appropriate. My support for publication in International Journal of Molecular Sciences is very strong.

 Response

We would like to thank the reviewer for the enthusiastic comments on our review, that were a boost of positive energy for us.

Reviewer 2 Report

The authors present a manuscript for the special issue:

"Molecular Mechanisms of Synaptic Plasticity: Dynamic Changes in Neurons Functions"

It is a timely and interesting review which however needs some refinements in my opinion.

Paragraph 2.1. should be revised. Firstly, it should be reasoned, why is the focus laid on 5-HT7R! This is not so clearly delineated in paragraph. It is mentioned that studies emphasise its role in a wide range of physiological functions in the CNS, and as an example the modulation of 5HT7R expression by miR-29a is given…this seems displaced. Then the authors turn to its wide expression across neuronal tissues. However, I think it is worth to mention and summarise at the beginning of this paragraph why this review is focussed on 5-HT7R. 

Afterwards, the authors claim that „It is noteworthy that, in mammals, this receptor exhibits a number of functional splice variants due 129 to the presence of introns in the 5-HT7R gene [38].“ However, this sentence stands alone and here functional implications (functional diversity?) should be discussed more. This should be better embedded and maybe a better link to the subsequent signalling can be done. The whole paragraph appears constructed and without a flow.

A Figure for paragraph 2.would be beneficial (summary of its function in the developing and adult brain)

Line 250 (paragraph 3)

„Experiments on animal models indicate that chronic treatment with lurasidone enhances 5-HT transmission in dorsal raphe nuclei by coordinated 5-HT1AR agonism and 5-HT7R antagonism through modulation of GABAergic and glutamatergic pathways, thus contributing to the augmentation of drug's antidepressive effects“

To understand this crucial aspect better, the crosstalk of HT signaling with GABAergic and glutamatergic pathways could be introduced in the introduction.

Paragraph 3 collect a bunch of important information, here it would be beneficial to collect everything in a table, to provide a better overview.

Paragraph 4 is important, as such a connection is usually poorly addressed in neuroscientific reviews. However, the authors switch between intestinal inflammation, and in immune cell activation, for which it is hard to follow. As this article belongs to a special issue  addressing "Molecular Mechanisms of Synaptic Plasticity: Dynamic Changes in Neurons Functions“, I assume that the majority of readers might lack essential background knowledge to fully understand the addressed issues in its current presentation. So here the authors need to revise this paragraph, and provide a figure that supports the readers´ understanding about the complex interaction between gut, immune cells and brain. 

Author Response

Response to Reviewer 2 Comments

It is a timely and interesting review which however needs some refinements in my opinion.

Point 1:Paragraph 2.1. should be revised. Firstly, it should be reasoned, why is the focus laid on 5-HT7R! This is not so clearly delineated in paragraph. It is mentioned that studies emphasise its role in a wide range of physiological functions in the CNS, and as an example the modulation of 5HT7R expression by miR-29a is given…this seems displaced. Then the authors turn to its wide expression across neuronal tissues. However, I think it is worth to mention and summarise at the beginning of this paragraph why this review is focussed on 5-HT7R.

Response 1:We mentioned and summarized at the beginning of  the paragraph why our review was focused on the 5-HT7R (lines 127-136).

The sentence related to miR-29a was actually displaced at the beginning of paragraph 2.1 and was removed from this section.

Point 2: Afterwards, the authors claim that „It is noteworthy that, in mammals, this receptor exhibits a number of functional splice variants due 129 to the presence of introns in the 5-HT7R gene [38].“ However, this sentence stands alone and here functional implications (functional diversity?) should be discussed more. This should be better embedded and maybe a better link to the subsequent signalling can be done. The whole paragraph appears constructed and without a flow.

Response 2: We added a few sentences (lines 142-147) to discuss the issue of  the splice variants of the 5-HT7R gene.

We also rearranged the signal transduction pathway paragraph, to make it more fluent (lines 148-155).

Point 3: A Figure for paragraph 2.would be beneficial (summary of its function in the developing and adult brain)

Response 3: We added a Figure (Figure 1) that describes the main functions of the brain 5-HT7R during development and in the adulthood. This figure also summarizes the neuropsychiatric and neurodevelopmental diseases associated to 5-HT7R dysfunction and discussed in our review.

Point 4: Line 250 (paragraph 3)

Experiments on animal models indicate that chronic treatment with lurasidone enhances 5-HT transmission in dorsal raphe nuclei by coordinated 5-HT1AR agonism and 5-HT7R antagonism through modulation of GABAergic and glutamatergic pathways, thus contributing to the augmentation of drug's antidepressive effects“

To understand this crucial aspect better, the crosstalk of HT signaling with GABAergic and glutamatergic pathways could be introduced in the introduction.

Response 4: We added a few sentences on this important aspect at the end of paragraph 1.1 (lines 48-56).

Point 5: Paragraph 3 collect a bunch of important information, here it would be beneficial to collect everything in a table, to provide a better overview.

Response to point 5: Instead of adding a table, we summarized the brain diseases discussed in our review in a box of Figure 1.

Point 6: Paragraph 4 is important, as such a connection is usually poorly addressed in neuroscientific reviews. However, the authors switch between intestinal inflammation, and in immune cell activation, for which it is hard to follow. As this article belongs to a special issue addressing "Molecular Mechanisms of Synaptic Plasticity: Dynamic Changes in Neurons Functions“, I assume that the majority of readers might lack essential background knowledge to fully understand the addressed issues in its current presentation. So here the authors need to revise this paragraph, and provide a figure that supports the readers´ understanding about the complex interaction between gut, immune cells and brain.

Response to point 6: The paragraph was thoroughly revised, as indicated by the track changes. A figure (Figure 2) was added to illustrate the influence of 5-HT7R in brain, gut and immune cells.

We would like to thank the reviewer for its constructive criticisms and insightful comments that helped us to improve our review.

Round 2

Reviewer 2 Report

The authors addressed most of the concerns of my initial review in the revised version. The table I suggested was optional. I think, the manuscript in its present form is a valuable contribution to the field, and I suggest the acceptance for publication in IJMS.

I found a minor spelling mistake in line 366 (lymphocytes). I suggest detailed spell check in the proofs.